# The Autophagy Inhibitor Bafilomycin Inhibits Antibody-Dependent Natural Killer Cell-Mediated Killing of Breast Carcinoma Cells

**DOI:** 10.3390/ijms26136273

**Published:** 2025-06-28

**Authors:** Ákos M. Bede, Csongor Váróczy, Zsuzsanna Polgár, Gergő Fazekas, Csaba Hegedűs, Endre Kókai, Katalin Kovács, László Virág

**Affiliations:** 1Department of Medical Chemistry, Faculty of Medicine, University of Debrecen, Egyetem Tér 1, 4032 Debrecen, Hungary; bedeakos1000@gmail.com (Á.M.B.); csongor.varoczy@gmail.com (C.V.); polgar.zsuzsanna@med.unideb.hu (Z.P.); fazekas.gergo@med.unideb.hu (G.F.); hcsaba@med.unideb.hu (C.H.); ekokai@med.unideb.hu (E.K.); 2National Academy of Scientist Education, Egyetem Tér 1, 4032 Debrecen, Hungary; 3Division of Dental Biochemistry, Department of Basic Medical Sciences, Faculty of Dentistry, University of Debrecen, Egyetem Tér 1, 4032 Debrecen, Hungary; 4HUN-REN-DE (Hungarian Reasearch Network—University of Debrecen) Cell Biology and Signaling Research Group, Egyetem Tér 1, 4032 Debrecen, Hungary

**Keywords:** natural killer cell, trastuzumab, antibody dependent cell-mediated cytotoxicity, apoptosis, autophagy

## Abstract

The resistance of breast cancer cells to therapeutic antibodies such as anti-HER2 trastuzumab can be overcome by engaging natural killer (NK) cells for killing antibody-binding tumor cells via antibody-dependent cellular cytotoxicity (ADCC). Here, we investigated how autophagy modulation affects trastuzumab-mediated ADCC in HER2-positive JIMT1 breast cancer cells and NK cells. Autophagy inducers (rapamycin and resveratrol) had no significant impact, but the inhibitor bafilomycin nearly abolished ADCC. Protection occurred when either cancer or NK cells were pretreated, indicating dual effects. Bafilomycin reduced phosphatidylserine externalization, the loss of plasma membrane integrity, caspase-3/7 activity, and DNA fragmentation. It downregulated pro-apoptotic *BAK1* and *BAX* without altering *BCL-2*. Additionally, bafilomycin decreased HER2 surface expression, impairing trastuzumab binding, and modulated immune regulators (*STAT1*, *CD95*, and *PD-L1*) in NK and/or in the cancer cells. Bafilomycin disrupted HER2 trafficking and induced HER2 internalization, leading to its accumulation in cytoplasmic vesicles. These findings show that autophagy inhibition by bafilomycin confers ADCC resistance by altering apoptosis, immune signaling, and HER2 dynamics. The study underscores autophagy’s role in antibody-based cancer therapy efficacy.

## 1. Introduction

In women, breast cancer is the most common type of malignant tumor, with 1–2% of cases occurring in men [1]. While ductal carcinoma, accounting for 80% of cases, forms in the breast’s milk ducts, lobular carcinoma (~10% of cases) arises from lobular cells of the milk-producing glands [2]. The World Health Organization estimates that this disease takes the lives of over 700,000 women annually and results in more than 2 million new cases being diagnosed [3]. Standard treatment approaches include the surgical excision of the tumor, radiotherapy, and cytotoxic chemotherapy. Additionally, treatment options like hormone therapy and monoclonal antibody therapy can enhance outcomes, based on the tumor’s cancer marker profile, particularly regarding hormone (estrogen and progesterone) receptor expression and the overexpression of the *HER2* oncogene [3,4,5,6,7,8].

The importance of tumor stromal cells in determining tumor behavior, tumor cell proliferation, and treatment resistance is becoming more and more obvious [9]. Important elements of the tumor stroma and different immune cells from the innate and adaptive immune systems can influence disease outcomes in both positive and negative ways [10,11,12,13]. The innate immune response, which serves as the body’s first line of defense, includes macrophages, dendritic cells, granulocytes, mast cells, NK cells, and NKT cells. In contrast, the adaptive immune system features CD8+ T lymphocytes, B-lymphocytes, and CD4+ T cells that help regulate both aspects of adaptive immunity, forming a critical part of the immune network associated with tumors [14,15,16].

By facilitating the destruction of cancer cells via immune effector cells, antibody-dependent cell-mediated cytotoxicity (ADCC) is essential to the antitumor immune response [17]. The main mediators of this process are antibodies, which attach to tumor antigens and label them for immune cells like macrophages and natural killer (NK) cells to recognize and eliminate. NK cells, for instance, have been shown to destroy breast cancer cells by ADCC when the therapeutic antibody trastuzumab binds to the tumor cells’ surface epidermal growth factor receptor HER2 [18].

Enhancing the direct or ADCC-mediated tumor cell killing of immune cells could provide therapeutic benefit for cancer patients. Despite the multiple mechanisms of action, some tumors show primary or acquired resistance to trastuzumab therapy, which is not related to decreased HER2 expression in tumor cells. Therefore, several novel strategies have been explored to enhance ADCC. These strategies include the supplementation of cytokines, manipulation of the effector cells, and modification of the therapeutic antibodies [19]. ADCC was shown to be enhanced by altering the Fc portion of the mAb by site-directed mutagenesis [20], changing the glycosylation of the Fc domain [21,22], or removing Fc domain fucosylation [23]. Additionally, asymmetrical engineering of the Fc portion [24] was also demonstrated to increase binding affinity to the activating FcγRIIIA. NK cell engagers have been designed for enhanced and prolonged NK cell-mediated responses by targeting different activating NK cell receptors [25]. Novel engagers include tetravalent, bispecific innate cell engagers, bi-/trispecific NK cell engagers, and multi-specific antibody-based constructs. A novel approach is nanoliposomes loaded with immunopotentiators that enhance the ADCC effect in HER2-positive breast cancer [26,27]. Here, we set out to investigate if autophagy modulators could modify ADCC efficiency. Autophagy is a cellular degradation and recycling process that plays a dual role in cancer progression and therapy resistance. It can act as a tumor suppressor by removing damaged cellular components and as a survival mechanism that encourages tumor growth under stressful circumstances [28]. This duality is crucial for understanding how autophagy contributes to cancer progression and therapy resistance. In our present study, we aimed to investigate the effect of autophagy modulator compounds in ADCC.

## 2. Results

### 2.1. Bafilomycin A1 Inhibits Antibody-Dependent Cellular Cytotoxicity

First, we set up an in vitro ADCC model as previously described. In this model, HER2+ human JIMT1 breast carcinoma cells were treated with the anti-HER2 monoclonal antibody trastuzumab and co-incubated with NK cells. Trastuzumab bridges the two cell types, triggering ADCC.

To evaluate the role of autophagy in ADCC, we tested the effects of the autophagy modulator compounds rapamycin, bafilomycin, and resveratrol. While rapamycin and resveratrol induce autophagy, bafilomycin inhibits the process as summarized in Table 1.

In accordance with our previous findings [29,30,31], we observed that ADCC was very effective in our model, resulting in >50% target cell death in 3 h (Figure 1A,B). Moreover, we found that while the autophagy inducers rapamycin and resveratrol had no effect on ADCC, the autophagy inhibitor bafilomycin provided almost complete protection from ADCC-mediated breast cancer cell death (Figure 1). This effect could be observed in two different setups, i.e., when only the cancer cells but not the NK cells or when only the NK cells but not the cancer cells were pretreated with the drugs (Figure 1A,B). In addition to the visual inspection of microscopic pictures, images were also quantitatively analyzed by determining the microplate surface area occupied by the JIMT1 cancer cells. From this experiment, we concluded that autophagy induction by rapamycin and resveratrol has no effect on either the ADCC co-cultures or on any of their cellular components. Bafilomycin, on the other hand, appeared to exert its ADCC inhibitory effect by acting both on the effector and the target cells. It is important to note that both rapamycin and bafilomycin A1 are nontoxic to JIMT1 and NK cells at the concentrations used. In contrast, resveratrol caused a slight reduction in cell numbers after 24 h (Appendix A).

### 2.2. Bafilomycin A1 Decreases Annexin V Binding, Csapase 3/7 Activity and Membrane Permeability in JIMT1 Cells During ADCC

ADCC-mediated cell death is a complex cell death modality dominated by apoptotic and necroptotic features [32,33]. We set out to investigate how bafilomycin affects various cell death parameters. First, we stained JIMT1 cultures (with or without bafilomycin treatment) with cell tracker blue to visualize cells. We also performed Annexin V staining to detect phosphatidylserine externalization (from the inner to the outer layer of the plasma membrane), which is a common feature in apoptotic cell death. Our data show that Annexin V staining is reduced in bafilomycin-treated JIMT1 cells (Figure 2A,B). Similarly, bafilomycin also reduced SYTOX green staining of the cells, which is used as a measure of plasma membrane integrity (a sign of necroptosis) (Figure 2A,B). In addition to immunofluorescent staining, we also measured the activity of apoptosis-executing enzymes caspase-3/7. Bafilomycin treatment significantly decreased caspase activity in JIMT1 cells in ADCC (Figure 3A).

### 2.3. Bafilomycin A1 Modifies the Expression of BCL-2 Family Genes in JIMT1 Cells and Reduces DNA Fragmentation in ADCC

Since apoptosis is the predominant cell death modality triggered by ADCC [34], we hypothesized that bafilomycin may interfere with key apoptosis regulatory mechanisms. Caspase-3 is the main executioner enzyme in apoptosis, and caspase-3 activity was reduced in bafilomycin-treated JIMT1 cells (Figure 3A). We determined the expression levels of *BAK*, *BAX*, and *BCL-2* genes. BAK and BAX are pro-apoptotic proteins. They promote apoptosis by permeabilizing the mitochondrial outer membrane, leading to the release of cytochrome c and other pro-apoptotic factors that activate caspases and initiate cell death. On the other hand, BCL-2 is an anti-apoptotic protein. It inhibits apoptosis by binding to and neutralizing BAK and BAX, preventing mitochondrial outer membrane permeabilization and thus blocking the release of cytochrome c. The balance between these proteins determines cell fate: when BAK and BAX activity outweighs BCL-2 inhibition, apoptosis is triggered; when BCL-2 dominates, cell survival is promoted. Their interplay is crucial for maintaining cellular homeostasis and responding to stress or damage.

We found that bafilomycin reduced *BAK1* and *BAX* expression while *BCL-2* expression was left unaffected (Figure 3B). The main mechanism in ADCC is the release of perforins and granzymes from effector cell granules. Perforins create pores in the cell membrane that facilitate granzyme B entry into the target cell, resulting in DNA fragmentation and apoptosis. DNA strand breaks detected in JIMT1 cells after 3 h of ADCC using the indirect TUNEL method are significantly reduced by bafilomycin pretreatment (Figure 3C).

### 2.4. Bafilomycin A1 Decreases Cell Surface HER2 Expression and Trastuzumab Binding in JIMT1 Cells

To further investigate the mechanism of bafilomycin-induced ADCC resistance, we determined the expression levels of HER2 in JIMT1 cells. We hypothesized that bafilomycin may reduce surface expression of HER2, the target protein of trastuzumab. Indeed, our flow cytometry data indicate that—as a measure of HER2 expression—trastuzumab binding was reduced in bafilomycin-treated cells (Figure 4A). The effect required a long (24 h) incubation but also could be observed at the earlier (3 h) timepoint. Binding of the antibody to NK cells, however, was unaffected by the autophagy inhibitor compound (Figure 4B).

### 2.5. Bafilomycin A1 Modulates the Expression of Genes Involved in ADCC

ADCC involves the activation of NK cells by antibodies that bind to Fc receptors. The most characterized Fc receptor on the NK cell membrane is CD16 or FcƴRIII [35]. In addition to HER2 and CD16, several other proteins (e.g., immune checkpoint proteins PD-1 and its partner PD-1L, CD95 a.k.a. Fas receptor and its ligand CD95L a.k.a. Fas ligand) are involved in NK cell cancer cell interactions [36]. Death receptor-induced apoptosis is a perforin-independent mechanism by which NK cells lyse target cells. Fas (CD95) is expressed on a wide variety of tissues, but Fas ligand (CD95L) expression is restricted to activated NK cells and cytotoxic T lymphocytes. Fas cross-linking induces changes such as membrane blebbing, nuclear condensation, and caspase activation. Fas is downregulated in a variety of cancers during tumor progression [37,38]. Moreover, the Signal Transducer and Activator of Transcription 1 (STAT1) protein is also crucial in the context of trastuzumab-mediated ADCC as it enhances the cytotoxic activity of NK cells through interferon signaling. By promoting the transcription of genes involved in NK cell function and upregulating effector molecules, STAT1 helps to amplify the immune response against HER2-positive cancer cells, thereby contributing to the therapeutic efficacy of trastuzumab. Therefore, we investigated whether bafilomycin has any effect on the expression of these key-tumor immunity-regulating proteins. Our results show that expression of *STAT-1* was reduced in the NK cells (Figure 5). In JIMT-1 cells, bafilomycin suppressed *STAT1*, *PD-1L*, and *CD95* expression (Figure 5).

### 2.6. Bafilomycin A1 Reduces HER2 Surface Expression in JIMT1 Cells

Intracellular trafficking plays a crucial role in the cell surface appearance of cell membrane proteins, such as HER2. This process ensures that proteins like HER2 are synthesized, modified, sorted, and transported to their correct locations within the cell, including the plasma membrane.

Although autophagy primarily targets damaged organelles and misfolded proteins, it can also affect the trafficking and localization of cell membrane proteins. Thus, we hypothesized that the perturbed trafficking of HER2 may be one of the molecular mechanisms underlying the cytoprotective effect of bafilomycin in our ADCC model. Indeed, our data show that the cell surface expression of HER2 is reduced in bafilomycin-treated JIMT-1 cells (Figure 6) and the protein is retained in cytoplasmic vesicles (Figure 6). Quantitative high-content analysis of fluorescent HER2 images confirmed this observation (Figure 6).

### 2.7. Bafilomycin A1 Induces HER-2 Internalization in JIMT1 Cells

Besides disrupted intracellular trafficking, receptor internalization likely plays a significant role in the reduced cell surface expression of the HER2 receptor and its accumulation in cytoplasmic vesicles. To investigate how bafilomycin affects receptor internalization, we set up a live-cell assay using the fluorescently labeled HER2 antibody. Our results demonstrate that HER2 internalization occurred within 3 h of bafilomycin treatment. By 24 h, anti-HER2 was almost completely internalized in bafilomycin-treated cells, while internalization could not be observed in control cells (Figure 7).

## 3. Discussion

Cancer immunotherapies revolutionized oncology, providing hope for patients with chemo- and/or radiation-resistant cancers. These immunotherapies include immune checkpoint inhibitors (e.g., PD-1/PD-L1, CTLA-4 inhibitors) that enhance T-cell activity, cancer vaccines (e.g., HPV vaccine, personalized neoantigen vaccines) that stimulate immune responses, and cytokine therapy (e.g., IL-2, IFN-α) that boosts immune cell function or the use of genetically modified T cells expressing chimeric antigen receptors (CAR-T Cell Therapy) [39]. Moreover, anticancer antibodies such as bispecific antibodies engaging immune cells and cancer cells simultaneously and antibody–drug conjugates that deliver toxic agents directly to cancer cells (e.g., trastuzumab emtansine) also represent powerful weapons in the anticancer armament [40]. In our study, we focused on the monoclonal antibody (trastuzumab) that targets HER2 in breast cancer. It has previously been shown that ADCC is a key effector mechanism in the therapeutic action of trastuzumab [41].

Using autophagy modulator compounds, we aimed to explore the role of autophagy in trastuzumab-mediated ADCC. Autophagy has been implicated in tumor progression and therapy resistance. Autophagy supports tumor growth by providing energy and nutrients, especially under conditions of metabolic stress, which are common in rapidly growing tumors [42]. The process helps maintain cellular homeostasis and prevents the accumulation of damaged proteins and organelles, which could otherwise lead to genomic instability and cancer progression [43]. Cancer cells exploit autophagy to survive chemotherapy and other treatments by managing stress and maintaining energy production [42]. It has been suggested that targeting the autophagy pathway, including its regulators like SIRT1 and miRNAs, presents a promising strategy to overcome therapy resistance and improve cancer treatment outcomes [44,45]. Inhibiting autophagy has shown potential in preclinical studies to restore chemotherapeutic sensitivity and increase cancer cell mortality [42].

Most studies have focused on the role of autophagy in chemosensitivity and chemoresistance, while relatively little attention has been paid to the role of autophagy in immunotherapies. Investigations into the role of autophagy in the antitumor actions of T cells suggested that inhibiting autophagy in tumor cells can sensitize them to T cell-mediated killing and enhance the efficacy of immunotherapies. On the other hand, promoting autophagy in T cells can improve their anti-tumor activity and persistence, potentially enhancing adoptive T-cell therapies (e.g., CAR-T cells). Our findings with rapamycin and resveratrol suggest that inducing autophagy may not affect the ADCC sensitivity of HER2+ breast cancer cells. Somewhat surprisingly, the autophagy inhibitor compound bafilomycin conferred almost complete protection to JIMT1 cells from trastuzumab and NK cell-mediated ADCC.

Cancer cell killing by ADCC relies mostly on the apoptotic pathway triggered by granzyme release from NK cells [34]. Apoptotic parameters (phosphatidylserine externalization and caspase activity) were decreased by bafilomycin, suggesting that the compound interferes with this cell death route. Activated NK cells also release perforins, which create pores in the membranes of cancer cells, leading to cell lysis and death by disrupting osmotic balance and facilitating the entry of pro-apoptotic molecules. Permeabilization of the target cell membrane was indicated by the uptake of the cell-impermeable SYTOX Green dye, and this cell death parameter was also decreased by bafilomycin. Fas-FasL binding also plays a role in ADCC by contributing to the cytotoxic mechanisms of NK cells [46]. When NK cells recognize the antibody-coated cells via Fc receptors, NK cells can induce apoptosis through Fas–FasL interactions. FasL on the NK cells binds to Fas receptors on target cells, triggering the extrinsic apoptotic pathway. This enhances the overall cytotoxic effect of ADCC, complementing other mechanisms like perforin/granzyme release.

The question arises as to the mechanism by which bafilomycin might inhibit ADCC. In many cases, tumor cells manage to survive chemo- and/or radiotherapy by activating autophagy. Based on these observations, it has been proposed that autophagy inhibition may sensitize cancer cells to these therapies. The autophagy inhibitor compound bafilomycin, however, provided protection to JIMT1 tumor cells from trastuzumab-induced NK cell-dependent cytotoxicity. These observations suggest that in this model, autophagy supports the apoptotic machinery of the target cells. One possible mechanism for the death-promoting role of autophagy may involve Fas ligand signaling. It has been shown that cells with a high autophagic flux are more sensitive to Fas ligand-induced apoptosis [47]. Therefore, in such situations, autophagy inhibition can be cytoprotective. Of note, Fas receptor expression was significantly lower in bafilomycin-treated JIMT1 cells, providing support for this hypothesis. A protein that could possibly link autophagy and Fas-mediated apoptosis could be the protein phosphatase Fap-1, whose degradation by autophagy enhances Fas-mediated apoptotic signaling [47].

The autophagy inhibitor compound bafilomycin also affected the expression of many genes involved in the regulation of cell death. For example, the expression of *STAT-1* was reduced in the NK cells, while in JIMT-1 cells, bafilomycin suppressed *BAK1*, *BAX*, *STAT1*, and *PD-1L* expression (Figure 3 and Figure 5). Several mechanisms may be involved in the crosstalk between gene expression and autophagy [48,49]. These include the stability of transcription factors (e.g., transcription factor EB, FOXO, or NF-κB), the modulation of epigenetic mechanisms, the degradation of signaling molecules (e.g., p62 or inhibitor of kappa B kinase), the activation of stress response pathways (e.g., the unfolded protein response), and many others [49,50,51,52,53,54]. Whether or not and how these effects of bafilomycin on gene expression contribute to autophagy/cell death inhibition requires further investigation.

A key mechanism underlying the tumor cell-protecting effect of bafilomycin in our ADCC model may be the reduction in cell-surface HER2 expression. In bafilomycin-treated JIMT1 cells, HER2 could be found in cytoplasmic vesicles, indicating impaired trafficking to the plasma membrane. Several mechanisms have previously been put forward to explain how autophagy can interfere with the trafficking of receptors to the membrane. These include the direct degradation of receptors, the degradation of trafficking machinery, altered endosomal sorting, the disruption of membrane dynamics, the regulation of signaling pathways, and selective autophagy of receptor complexes [55,56]. Moreover, as an inhibitor of the vacuolar proton pump, bafilomycin may also directly block trafficking through early and late endosomes [57]. Determining which of these possible scenarios explains our findings requires further investigation.

The effectiveness of therapeutic antibodies and antibody–drug conjugates largely depends on receptor internalization. The overexpression of HER2 by certain tumor cells can provide therapeutic advantages, but receptor internalization and lysosomal trafficking significantly influence the efficiency of various therapeutic approaches. Different HER2 antibodies have been shown to internalize with varying efficiencies. For instance, trastuzumab, the standard treatment for HER2-positive breast cancer, was shown to induce HER2 internalization to a small extent and with very slow kinetics [58]. In contrast, polyclonal anti-HER2 antibodies and bispecific antibodies facilitate rapid receptor internalization and degradation [58,59]. Our study indicates that treatment with bafilomycin significantly enhances HER2 internalization, which might slow down tumor growth. However, it also significantly decreases the efficiency of antibody-based treatments such as antibody-dependent cellular cytotoxicity. Our findings are in line with previous studies where bafilomycin was shown to interfere with transferrin receptor externalization [60].

While autophagy contributes to cancer progression and therapy resistance, it also offers potential therapeutic targets. The complexity of its regulatory networks, involving miRNAs, m6A modifications, and proteins like SIRT1, underscores the need for further research to develop effective cancer therapies that manipulate autophagy pathways. Our present findings highlight the importance of testing the effects of autophagy modulator drugs in disease- and therapy-specific research models to identify the model-specific roles of autophagy inducers and inhibitors. Based on our findings, bafilomycin significantly increases the rate of HER2 receptor internalization and may alter intracellular trafficking, ultimately leading to a disruption of receptor turnover in JIMT1 cells. Although our study primarily focused on the impact of bafilomycin on JIMT1 cells, gene expression analyses revealed the intriguing possibility that bafilomycin A1 may also compromise the functionality of natural killer (NK) cells. Notably, we observed a marked reduction in *STAT1* expression levels within NK cells, suggesting that bafilomycin disrupts the mechanisms underlying NK cell-mediated tumor surveillance. This interference could hinder the ability of NK cells to effectively identify and eliminate tumor cells, thereby impacting overall immune responses.

Our in vitro model demonstrated that bafilomycin A1 induces the internalization of HER2 in JIMT1 breast cancer cells. Since bafilomycin A1 inhibits lysosomal acidification, it prevents the degradation of internalized HER2. We hypothesize that the internalized HER2 receptors accumulate in recycling endosomes, as bafilomycin A1 may interfere with the normal endosomal sorting mechanisms, potentially leading to the impaired recycling of HER2 to the surface of JIMT1 cells. These findings need to be validated using patient-derived xenografts or in vivo models. This validation is essential to predict resistance to HER2-targeted therapies, assess treatment responsiveness, select suitable treatment combinations, and ultimately develop more effective HER2-targeted therapies. Considering the effects on HER2 receptor internalization and its degradation, bafilomycin A1 can potentially contribute to acquired trastuzumab resistance of breast cancer cells. Bafilomycin treatment may lead to HER2 accumulation in locations that are less accessible to trastuzumab, reducing its efficacy.

As autophagy inhibitors are currently being investigated in anticancer clinical trials, it is important to consider how they may affect other therapeutic approaches. Our data indicate that the autophagy inhibitor bafilomycin A1 could interfere with immunotherapies by reducing the availability of HER2 on the cell surface. This reduction may limit the binding of trastuzumab and hinder essential immune mechanisms. Therefore, treatment strategies that involve bafilomycin A1 or other V-ATPase inhibitors should be approached with caution, as they may disrupt critical functions of HER2-targeted immunotherapy.

## 4. Materials and Methods

### 4.1. Cell Lines

JIMT-1 breast cancer cells were cultured in DMEM/F-12 medium (L0092, Biowest, Bradenton, FL, USA) supplemented with 10% fetal bovine serum (FB-1090, Biosera, Cholet, France), 10 μg/mL insulin (HI0219, Humulin R 100 NE/mL, Eli Lilly Nederland B.V., Utrecht, The Netherlands), 1% L-glutamine (GLN-B, Capricorn Scientific, Ebsdorfergrund, Germany), and 1% penicillin-streptomycin (LM-A4118, Biosera, France). The CD16.176 V.NK-92 cell line was a kind gift from Prof. Dr. György Vereb (Department of Biophysics and Cell Biology, Debrecen, Hungary). NK cells were maintained in MEM Alpha (LM-E1148, Biosera, France) supplemented with 20% FBS, 1% MEM-NEAA (11140-050, Gibco, Jenks, OK, USA), 1% odium pyruvate (NPY-B, Capricorn Scientific, Germany), 1% L-glutamine, 1% penicillin-streptomycin, and 20 ng/mL human IL-2 (SRP3085, Sigma-Aldrich, Taufkirchen, Germany). Both cell lines were cultured at 37 °C in a humidified atmosphere containing 5% CO_2_ and were routinely checked for the absence of mycoplasma contamination.

### 4.2. Live Cell Imaging in ADCC

Our high-content screening (HCS)-based assay protocol is discussed in our previous publications [29,30,31]. Briefly, JIMT1 cells (10^4^ cells per well) were seeded in 96-well HCS plates (Cell Carrier Ultra HCS microplates, PerkinElmer, Waltham, MA, USA), while JIMT1 or NK cells were pretreated with 500 nM rapamycin (553210, Merck KGaA, Darmstadt, Germany), 500 nM bafilomycin A1 (11038, Cayman Chemical, Ann Arbor, MI, USA), or 50 μM resveratrol (70675, Cayman Chemical, Ann Arbor, MI, USA) for 20 h before inducing antibody-dependent cellular cytotoxicity (ADCC). DMSO (02610-101-340, Molar Chemicals, Halásztelek, Hungary) was used as a vehicle control. JIMT1 cells were stained with 1 μM Calcein-AM (206700, Merck KGaA, Germany) for 1 h. After washing JIMT1 cells with cell culture media, NK cells were added in the presence of 1 µg/mL trastuzumab (LM134405-CA001, Herzuma^®^, humanized anti-HER2 monoclonal antibody, Celltrion Healthcare Hungary, Budapest, Hungary) in a 2:1 effector-to-target ratio. Images were captured immediately after NK addition (0 h) and 3 h later with an Opera Phenix High-Content Analysis system (PerkinElmer, Waltham, MA, USA) using a 10× air objective in non-confocal mode. Images were captured in brightfield and fluorescence channels (ex 488/em 500–550 for Calcein-AM). To test the effect of each compound on cell viability, we established control groups without ADCC that were pretreated with rapamycin, bafilomycin, or resveratrol. The trastuzumab-containing NK medium was then added without NK cells. The ADCC efficiency and JIMT1 image region areas were determined after image analysis with the Harmony Analysis Software v.4.9 (Perkin Elmer). For all steps of the image analysis sequence, the built-in modules of the Harmony software were used. A total of 156 images were analyzed for each condition.

### 4.3. Annexin V—Alexa 647 and SYTOX Green Staining

JIMT1 cells (10^4^ cells per well) were seeded in 96-well HCS plates (Cell Carrier Ultra HCS microplates, PerkinElmer, Waltham, MA, USA) and pretreated with 500 nM bafilomycin A1 (11038, Cayman Chemical, Ann Arbor, MI, USA) or vehicle for 20 h before inducing antibody-dependent cellular cytotoxicity (ADCC). JIMT1 cells were stained with 10 μM CellTracker^TM^ Blue CMAC (C2110, Invitrogen, Life Technologies Corporation, Eugene, OR, USA) for 1 h. After washing JIMT1 cells with cell culture media, NK cells were added in the presence of 1 µg/mL trastuzumab (LM134405-CA001, Herzuma^®^, humanized anti-HER2 monoclonal antibody, Celltrion Healthcare Hungary) in a 2:1 effector-to-target ratio and the co-culture was incubated for 3 h. Cells were then incubated with Annexin V—Alexa Fluor 647^TM^ conjugate (in 1:100) (A23204, Invitrogen, Life Technologies Corporation, Eugene, OR, USA) and 5 μM SYTOX^TM^ Green Nucleic Acid Stain (S7020, Invitrogen, Life Technologies Corporation, Eugene, OR, USA), and images were captured with the Opera Phenix High-Content Analysis system (PerkinElmer, Waltham, MA, USA) using a 10× air objective in non-confocal mode (ex 405/em 435–480 for CellTracker Blue; ex 488/em 500–550 for SYTOX Green; ex 640/em 650–760 for Alexa 647). Images were analyzed with the Harmony software (Perkin Elmer) to select CellTracker Blue-stained JIMT1 cells and calculate Alexa 647 and SYTOX Green intensity. For all steps of the image analysis sequence, the built-in modules of the Harmony software were used. A total of 156 images were analyzed for each condition.

### 4.4. Caspase 3/7 Activity

JIMT1 cells (10^4^ cells per well) were seeded in 96-well HCS plates (Cell Carrier Ultra HCS microplates, PerkinElmer, Waltham, MA, USA) and pretreated with 500 nM bafilomycin A1 or vehicle for 20 h before inducing antibody-dependent cellular cytotoxicity (ADCC). JIMT1 cells were stained with 10 μM CellTracker^TM^ Blue CMAC (C2110, Invitrogen, Life Technologies Corporation, Eugene, OR, USA) for 1 h. After washing JIMT1 cells with cell culture media, NK cells were added in the presence of 1 µg/mL trastuzumab (LM134405-CA001, Herzuma^®^, humanized anti-HER2 monoclonal antibody, Celltrion Healthcare Hungary) in a 2:1 effector-to-target ratio. CellEvent^TM^ Caspase-3/7 Green (C10423, Invitrogen, Life Technologies Corporation, Eugene, OR, USA) fluorescent caspase substrate was added to the co-cultures in a 1:500 ratio at the beginning of ADCC. Images were captured after 3 h ADCC with the Opera Phenix High-Content Analysis system (PerkinElmer, Waltham, MA, USA) using a 10× air objective in non-confocal mode (ex 405/em 435–480 for CellTracker Blue; ex 488/em 500–550 for CellEvent Caspase-3/7). Images were analyzed with the Harmony software (Perkin Elmer) to select CellTracker Blue-stained JIMT1 cells and calculate intensity values for the fluorescent caspase substrate. For all steps of the image analysis sequence, the built-in modules of the Harmony software were used. A total of 156 images were analyzed for each condition.

### 4.5. RT-qPCR

Gene expression levels were analyzed by RT-qPCR. Cells were homogenized in Tri Reagent (TR118, Molecular Research Center, Cincinnati, OH, USA), while total RNA was isolated by the phenol/chloroform method. RNA concentration was measured using Nanodrop, and 2 μg of RNA from each sample was reverse-transcribed to cDNA with the High-Capacity cDNA Reverse Transcription Kit (4368813, Applied Biosystems, Thermo Fisher Scientific, Vilnius, Lithuania). All PCR reactions were carried out using the Xceed SyGreen Mix Lo-ROX (LPCR10605L, Institute of Applied Biotechnologies, Prague, Czech Republic) in a LightCycler© 480 II thermocycler (Roche Diagnostics, Mannheim, Germany) for 40 cycles. Relative gene expression was determined by the 2^−ΔΔCt^ method. Cyclophilin A and GAPDH were used as internal controls, and gene expression values were normalized to the geometric mean of the housekeeping genes. Sequence of gene-specific primers is given in Table 2.

### 4.6. Flow Cytometry Analysis of Trastuzumab Binding

JIMT1 and NK cells were treated with 500 nM bafilomycin A1 or the vehicle for the indicated time intervals. Following the trypsinization of JIMT1, cells were centrifuged and resuspended in cell culture media. Cells were incubated with Alexa 488-conjugated trastuzumab (kind gift from Prof. Dr. György Vereb, Department of Biophysics and Cell Biology, Debrecen, Hungary) or FITC-IgG1 κ isotype standard (1:1000) (03004C, Pharmingen, NJ, USA) for 1 h. The Alexa 488 intensity of cell-bound trastuzumab was analyzed by the Novocyte 3000 flow cytometer (Accela, Prague, Czech Republic).

### 4.7. Immunocytochemistry

JIMT1 cells (10^4^ cells per well) were seeded in 96-well HCS plates (Cell Carrier Ultra HCS microplates, PerkinElmer, Waltham, MA, USA) and treated with 500 nM bafilomycin A1 or the vehicle for 24 h. Cells were fixed with 4% formaldehyde (03300-101-340, Molar Chemicals, Hungary) for 15 min, permeabilized and blocked with 1% BSA (421501J, VWR Life Science, Leuven, Belgium), and dissolved in PBS containing 0.1% Triton X-100 (3.20150, Spektrum 3D, Debrecen, Hungary) for 1 h at room temperature. Cells were then incubated with the anti-human HER2/FITC antibody (BMS120FI, Invitrogen, Bender MedSystems GmbH, Vienna, Austria) diluted in PBS in a 1:100 ratio. Cell nuclei were then stained with 1 μg/mL DAPI (D-1306, Molecular Probes, Eugene, OR, USA), and F-actin was labelled with Texas Red-X Phalloidin (1:500) (T7471, Invitrogen, Life Technologies Corporation, Eugene, OR, USA). Images were captured with the Opera Phenix High-Content Analysis system (PerkinElmer, Waltham, MA, USA) using a 63× water objective in confocal mode (ex 405/em 435–480 for DAPI; ex 488/em 500–550 for HER2-FITC; ex 488/em 570–630 for Texas Red). Images were analyzed with the Harmony software (Perkin Elmer) to select JIMT1 cells and determine the membrane and cytoplasmic regions of cells. The distribution of HER2-FITC within the cells was evaluated, and the parameters for cytoplasmic green fluorescent plots were also determined. Nuclear morphology was evaluated based on DAPI staining. For all steps of the image analysis sequence, the built-in modules of the Harmony software were used. A total of 864 images were analyzed for each condition.

### 4.8. In Situ Apoptosis Detection by the Indirect TUNEL Method

JIMT1 cells (10^4^ cells per well) were seeded in 96-well HCS plates (Cell Carrier Ultra HCS microplates, PerkinElmer, Waltham, MA, USA) and treated with 500 nM bafilomycin A1 or the vehicle for 20 h before inducing antibody-dependent cellular cytotoxicity (ADCC). NK cells were then added in the presence of 1 µg/mL trastuzumab (LM134405-CA001, Herzuma^®^, humanized anti-HER2 monoclonal antibody, Celltrion Healthcare Hungary) in a 2:1 effector-to-target ratio, and the co-culture was incubated for 3 h. Apoptotic cells were detected by labeling DNA strand breaks using the indirect TUNEL method using the ApopTag^®^ Plus Peroxidase In Situ Apoptosis Kit (S7101, Sigma-Aldrich, Germany) according to the manufacturer’s instructions. Briefly, cells were first fixed in 1% paraformaldehyde in PBS, pH 7.4, for 10 min at RT, then washed twice with PBS and post-fixed in precooled ethanol–acetic acid 2:1 for 5 min at −20 °C. After washing twice with PBS, endogenous peroxidase activity was quenched with 3.0% hydrogen peroxide in PBS for 5 min at RT, followed by two washes with PBS and the addition of equilibration buffer. Cells were then incubated with the TdT (terminal deoxynucleotidyl transferase) enzyme at 37 °C for 1 h to label the free 3′OH DNA termini in situ with digoxigenin-conjugated nucleotides. The stop/wash buffer was added for 10 min to deactivate the enzyme and stop the reaction. After washing three times with PBS, the anti-digoxigenin peroxidase conjugate was added for 30 min at RT, and DNA fragments, which were labeled with the digoxigenin-nucleotide, were allowed to bind to an anti-digoxigenin antibody. Cells were washed four times with PBS before being incubated with DAB peroxidase substrate for 6 min at room temperature, followed by three washes with water. Brightfield images were captured using the Opera Phenix High-Content Analysis system (PerkinElmer, Waltham, MA, USA) with a 40× water objective in nonconfocal mode. Image analysis was conducted using ImageJ software (https://imagej.net; access date 26 November 2021). The total area of all cells and TUNEL-positive regions was determined based on the histogram of pixel intensities. Individual objects were identified by an automated thresholding procedure. The Minimum method was used to detect TUNEL-positive regions, and the Triangle method was used to identify cells. A total of 60 images were analyzed for each condition.

### 4.9. HER2 Internalization

A receptor-mediated antibody internalization live-cell assay was established to monitor HER2 turnover. JIMT1 cells (10^4^ cells per well) were cultured in 96-well HCS plates (Cell Carrier Ultra HCS microplates, PerkinElmer, Waltham, MA, USA) and stained with the DRAQ5 Fluorescent Probe (62251, Thermo Fisher Scientific, Waltham, MA USA) for 30 min at 37 °C. After washing twice with cell culture media, the FITC-labelled anti-HER2 antibody (BMS120FI, Invitrogen, Bender MedSystems GmbH, Austria) was added to the cells at a dilution of 1:100, either with 500 nM bafilomycin A1 or a vehicle, and incubated for 24 h. Confocal images were acquired using the Opera Phenix HCA equipment with a 63× water objective (ex 488/em 500–550 for HER2-FITC; ex 640/em 650–760 for DRAQ5) immediately after treatment, as well as 3 h and 24 h post-treatment. Images were analyzed with the Harmony software (Perkin Elmer) to identify JIMT1 cells and define the membrane and cytoplasmic regions. The distribution of HER2-FITC within the cells was evaluated, and parameters for cytoplasmic green fluorescent spots were also determined. A total of 444 images were analyzed for each condition.

### 4.10. Statistical Analysis

Statistical analysis was performed with GraphPad Prism 8.0.1 (GraphPad Software Inc., San Diego, CA, USA). Data were analyzed with One-way ANOVA. All plots represent three independent experiments, with values given as mean ± SD.

## Figures and Tables

**Figure 1 ijms-26-06273-f001:**
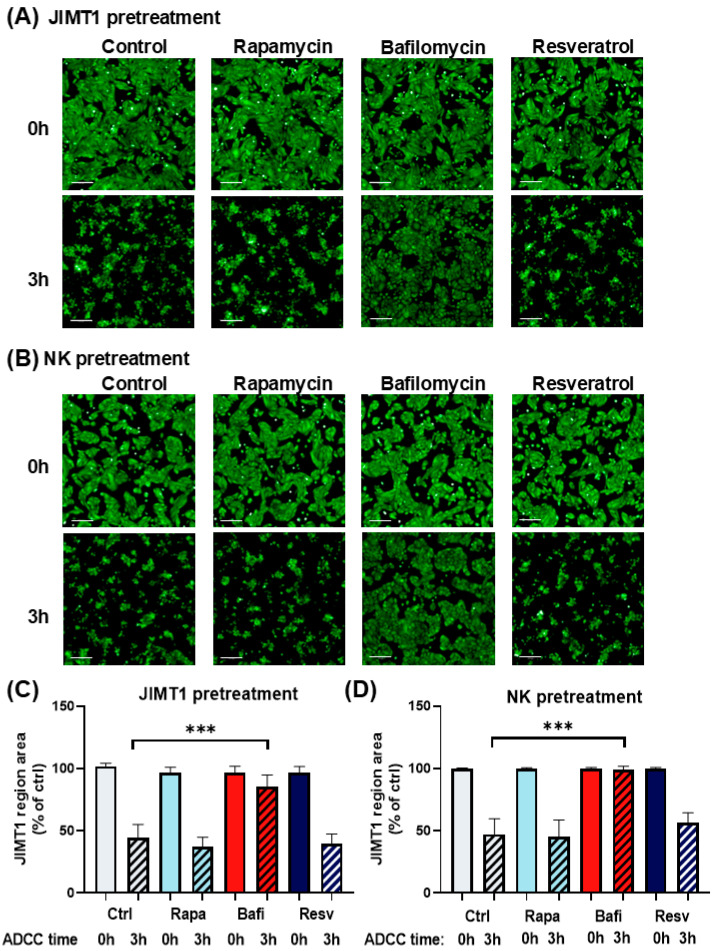
Bafilomycin A1 effectively inhibits antibody-dependent cellular cytotoxicity. Live cell imaging of JIMT1 cells in ADCC with NK92CD16 cells at a 2:1 effector–target ratio for 3 h. (**A**) JIMT1 cells in 96-well HCS plates were pretreated with 500 nM rapamycin, 500 nM bafilomycin A1, or 50 μM resveratrol for 20 h, then stained with Calcein-AM. NK cells were added in the presence of trastuzumab, and images were taken immediately after NK addition (0 h) and 3 h later. (Green: Calcein-AM staining of JIMT1 cells). (**B**) JIMT1 cells were seeded into 96-well HCS plates. NK92CD16 cells were pretreated with 500 nM rapamycin, 500 nM bafilomycin A1, or 50 μM resveratrol for 20 h. JIMT1 cells were stained with Calcein-AM before NK cells and trastuzumab were added, and images were taken immediately after NK addition (0 h) and 3 h later. (Green: Calcein-AM staining of JIMT1 cells). (**C**) JIMT1 cells were pretreated with autophagy modulators and cell confluence was determined by image analysis at 0 h and 3 h of ADCC. (**D**) NK cells were pretreated with autophagy modulators and JIMT1 cell confluence was determined by image analysis at 0 h and 3 h of ADCC. All plots represent at least three independent experiments; values are given as mean ± SD (*** *p* < 0.001) (scale bar is 200 µm).

**Figure 2 ijms-26-06273-f002:**
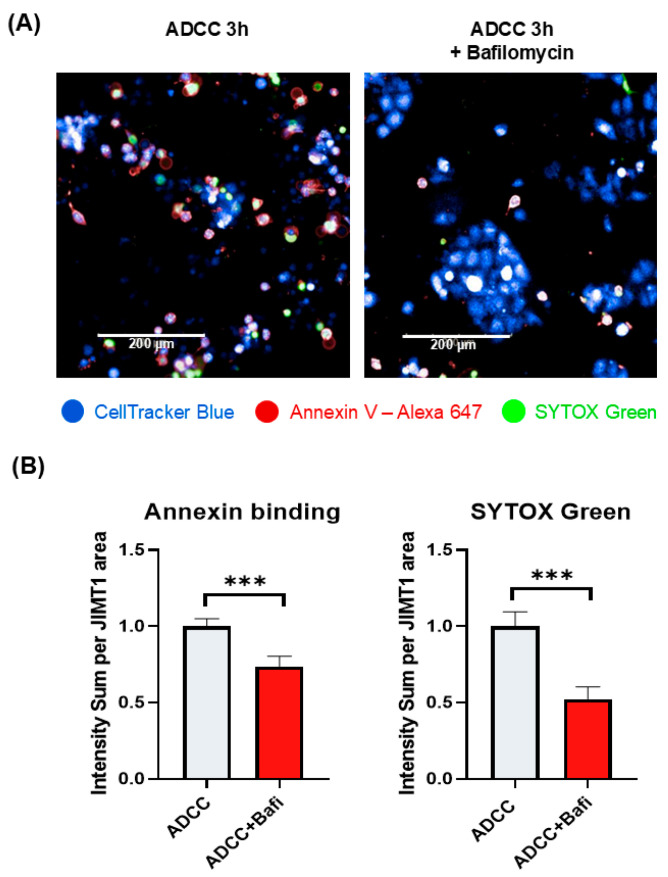
Bafilomycin A1 pretreatment of JIMT1 cells prevents apoptotic cell death and decreases membrane permeability in ADCC. (**A**) Annexin V—SYTOX Green staining of JIMT1 cells after 3 h ADCC. JIMT1 cells were stained with CellTracker Blue and pretreated with 500 nM bafilomycin A1 or vehicle for 20 h. After 3 h of ADCC, cells were stained with Annexin V—Alexa 647 (red) and SYTOX Green. (**B**) Image analysis and evaluation of Annexin V binding and SYTOX Green staining in JIMT1 cells. Alexa 647 and SYTOX Green intensity values are normalized to JIMT1 cell region area. All plots represent three independent experiments; values are given as mean ± SD (*** *p* < 0.001) (scale bar is 200 µm).

**Figure 3 ijms-26-06273-f003:**
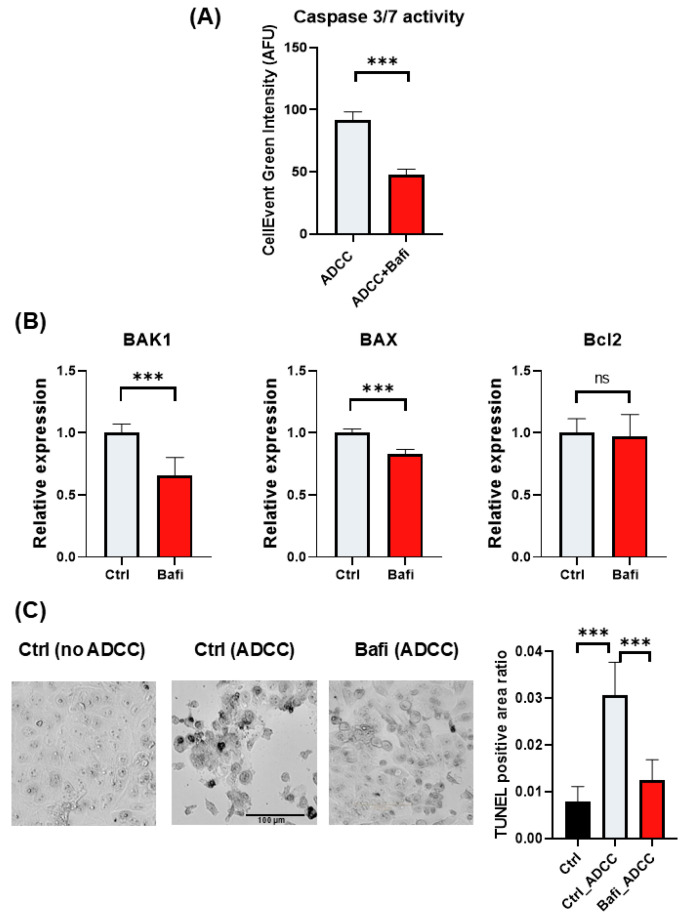
Effect of bafilomycin A1 on caspase activation and the expression of *BCL-2* family genes in JIMT1 cells. (**A**) Evaluation of caspase 3/7 activity by live cell imaging in JIMT1 cells during ADCC. JIMT1 cells in 96-well HCS plates were stained with CellTracker Blue and pretreated with 500 nM bafilomycin A1 or vehicle for 20 h. Apoptotic cells were labelled with CellEvent Caspase-3/7 Green and images were taken after 3 h ADCC. The intensity of the fluorescent caspase substrate was evaluated with image analysis. (**B**) JIMT1 cells were treated with 500 nM bafilomycin A1 or vehicle for 24 h. mRNA expression levels of *BCL-2* family genes in JIMT1 cells were determined by RT-qPCR. Expression levels were normalized to *cyclophilin A* and *GAPDH* and presented as fold change compared to control. All plots represent three independent experiments; values are given as mean ± SD (*** *p* < 0.001; ns = not significant). (**C**) DNA strand breaks were detected in ADCC cultures with indirect TUNEL method. JIMT1 cells were treated with 500 nM bafilomycin A1 or vehicle for 20 h before NK cells were added in the presence of trastuzumab for 3 h (at 2:1 effector–target ratio). TUNEL-positive area ratio was determined by image analysis. Plots represent three independent experiments; values are given as mean ± SD (*** *p* < 0.001) (scale bar is 100 µm).

**Figure 4 ijms-26-06273-f004:**
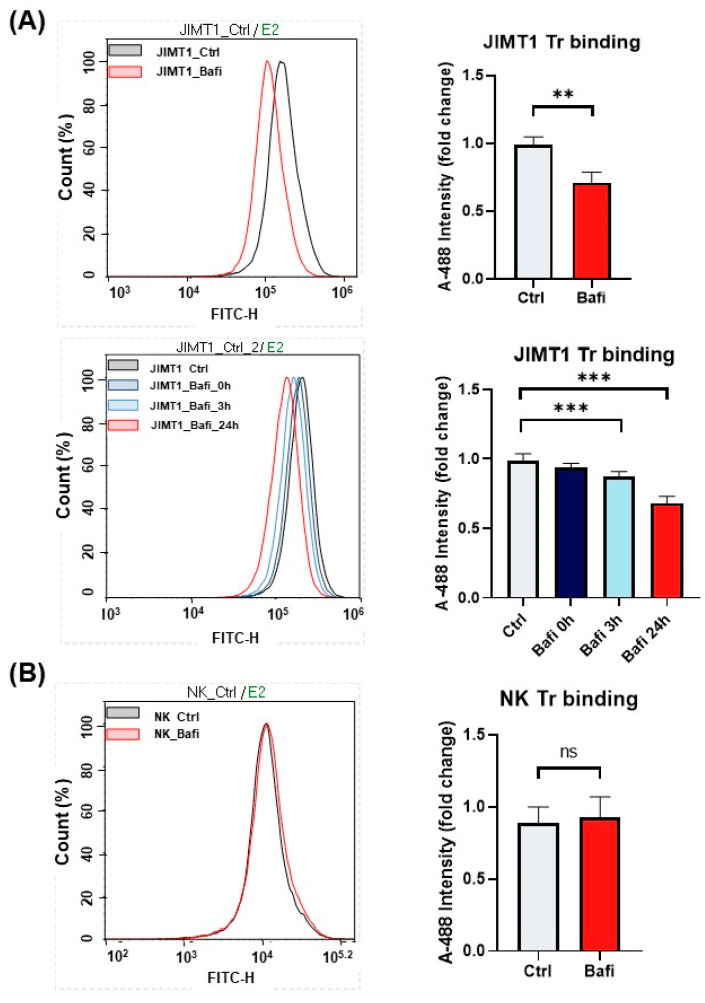
Bafilomycin A1 decreases cell surface HER2 expression and trastuzumab binding in JIMT1 cells. Trastuzumab (Tr) binding of JIMT1 and NK cells was analyzed by flow cytometry. (**A**) JIMT1 cells were treated with 500 nM bafilomycin A1 or vehicle for 24 h (upper) or for the indicated times (lower) and labelled with Alexa 488-conjugated trastuzumab. (**B**) NK cells were treated with 500 nM bafilomycin A1 or vehicle for 24 h and labelled with Alexa 488-conjugated trastuzumab. Alexa 488 intensity median values are presented as fold change compared to control. All plots represent three independent experiments; values are given as mean ± SD (** *p* < 0.01; *** *p* < 0.001; ns = not significant).

**Figure 5 ijms-26-06273-f005:**
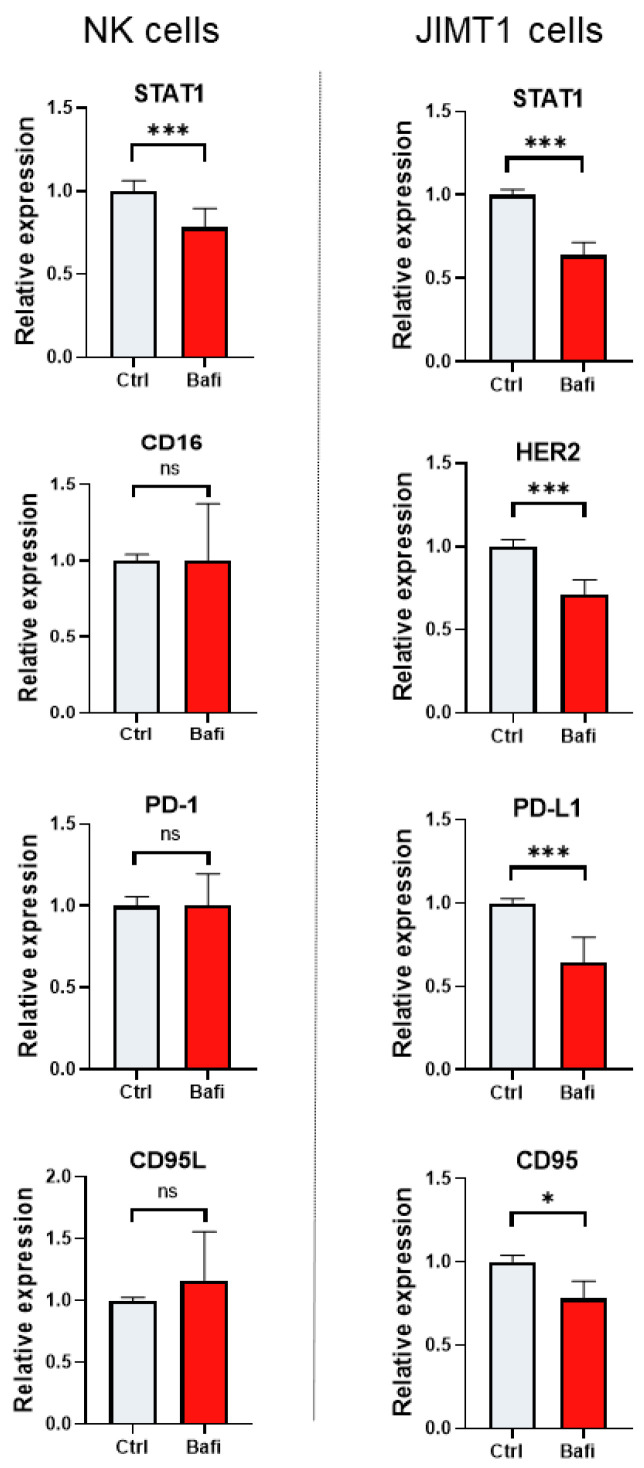
Bafilomycin A1 modulates the expression of genes involved in ADCC. JIMT1 and NK cells were treated with 500 nM bafilomycin A1 or vehicle for 24 h. mRNA levels of genes involved in NK cell function were determined by RT-qPCR. Expression levels were normalized to *cyclophilin A* and *GAPDH* and presented as fold change compared to control. All plots represent three independent experiments; values are given as mean ± SD. (* *p* < 0.05, *** *p* < 0.001, ns = not significant).

**Figure 6 ijms-26-06273-f006:**
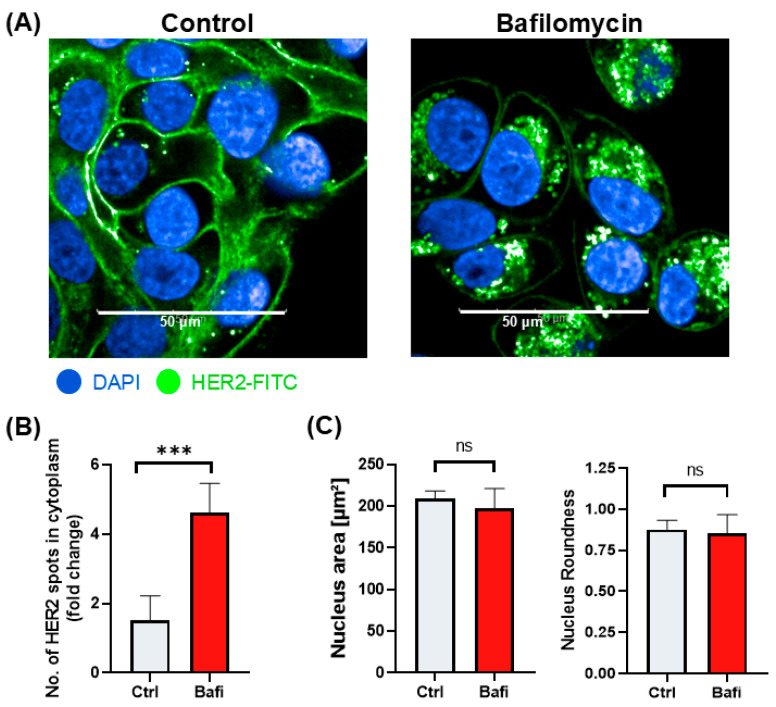
Bafilomycin A1 reduces HER2 surface expression by blocking HER2 intracellular trafficking. (**A**) Confocal microscopic images of HER2 staining in JIMT1 cells. JIMT1 cells in 96-well HCS plates were treated with 500 nM bafilomycin A1 or vehicle for 24 h. Cells were fixed and stained with FITC-labelled anti-HER2 antibody, and DAPI. Confocal images were taken with the Opera Phenix HCA equipment. (**B**) Image analysis and evaluation of HER2 localization in JIMT1 cells. FITC intensity values were analyzed in the membrane region and cytoplasmic region. The number of HER2+ cytoplasmic vesicles indicates altered intracellular trafficking. (**C**) Image analysis of nuclear morphology of JIMT1 cells. All plots represent three independent experiments; values are given as mean ± SD (*** *p* < 0.001; ns = not significant) (scale bar is 50 µm).

**Figure 7 ijms-26-06273-f007:**
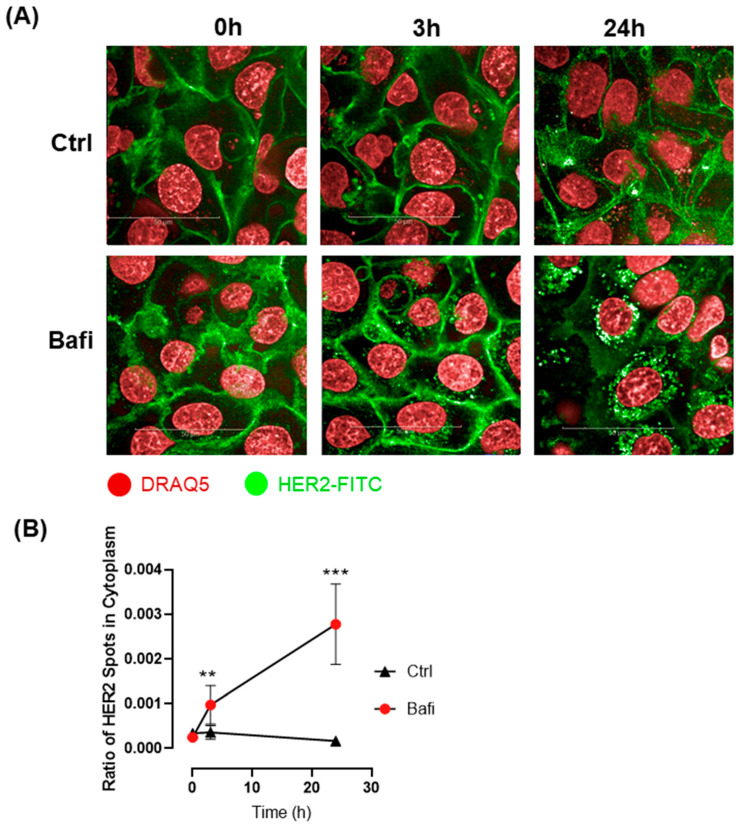
Bafilomycin A1 induces HER-2 internalization. (**A**) Confocal microscopic images were obtained to visualize HER2 live staining in JIMT1 cells. JIMT1 cells seeded in 96-well HCS plates were stained with DRAQ5 Fluorescent Probe. After washing, FITC-labelled anti-HER2 antibody was added to the cells, either with 500 nM bafilomycin A1 or vehicle and incubated for 24 h. Confocal images were taken using the Opera Phenix HCA equipment immediately after treatment, as well as 3 h and 24 h post-treatment. (**B**) Image analysis was used to evaluate HER2 localization in JIMT1 cells. FITC-stained spots were analyzed in the cytoplasmic region of cells. HER2-containing vesicles were observable after 3 h in bafilomycin A1-treated cells. Notably, 24 h bafilomycin A1 treatment dramatically increased the number of HER2-FITC-containing vesicles, indicating that HER2 was internalized. All plots represent three independent experiments; values are given as mean ± SD (** *p* < 0.01, *** *p* < 0.001) (scale bar is 50 µm).

**Table 1 ijms-26-06273-t001:** Pharmacological effects of autophagy modulator compounds used in this study.

Compound	Effect on Autophagy	Key Molecular Mechanism(s)
Rapamycin	Inducer	Inhibits mTORC1 → relieves inhibition of ULK1 → activates autophagy initiation.
Bafilomycin	Inhibitor	Inhibits V-ATPase → blocks lysosomal acidification and autophagosome-lysosome fusion.
Resveratrol	Inducer	Activates AMPK and SIRT1 → inhibits mTORC1 and deacetylates autophagy-related proteins.

**Table 2 ijms-26-06273-t002:** Sequence of the gene-specific primers.

	Forward Primer	Reverse Primer
BAK1	TTACCGCCATCAGCAGGAACAG	GGAACTCTGAGTCATAGCGTCG
BAX	TCAGGATGCGTCCACCAAGAAG	TGTGTCCACGGCGGCAATCATC
Bcl2	ATCGCCCTGTGGATGACTGAGT	GCCAGGAGAAATCAAACAGAGGC
CD16	GGTGACTTGTCCACTCCAGTGT	ACCATTGAGGCTCCAGGAACAC
CD95	GGACCCAGAATACCAAGTGCAG	GTTGCTGGTGAGTGTGCATTCC
CD95L	GGTTCTGGTTGCCTTGGTAGGA	CTGTGTGCATCTGGCTGGTAGA
HER2	GGAAGTACACGATGCGGAGACT	ACCTTCCTCAGCTCCGTCTCTT
PD-1	TGCCTGTGTTCTCTGTGGAC	GAGCAGTGTCCATCCTCAGG
PD-L1	GTTGAAGGACCAGCTCTCCC	TCCAGATGACTTCGGCCTTG
STAT1	ATGGCAGTCTGGCGGCTGAATT	CCAAACCAGGCTGGCACAATTG

## Data Availability

The data that support the findings of this study are available from the corresponding author [lvirag@med.unideb.hu] upon reasonable request.

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
