# Peer review of "The Autophagy Inhibitor Bafilomycin Inhibits Antibody-Dependent Natural Killer Cell-Mediated Killing of Breast Carcinoma Cells"

_ijms, 2025, doi:10.3390/ijms26136273_

Round 1
Reviewer 1 Report
Comments and Suggestions for Authors
Journal: IJMS (ISSN 1422-0067)
Manuscript ID: ijms-3529309
Type: Article
Title: The autophagy inhibitor bafilomycin inhibits antibody-dependent natural killer cell-mediated killing of breast carcinoma cells
Authors: László Virág * , Ákos Máté Bede , Csongor Váróczy , Zsuzsanna Polgár , Gergő Fazekas , Csaba Hegedűs , Endre Kókai , Katalin Kovács
Reviewer Comments:
The author has submitted a research manuscript proposing that Bafilomycin A1 inhibits the anticancer effect of Trastuzumab, a therapeutic agent targeting Human Epidermal Growth Factor Receptor 2 (HER2/ERBB2) in breast cancer. The hypothesized mechanism suggests that Bafilomycin A1 binds to HER2 on the breast cancer cell membrane, subsequently leading to its internalization and resulting in drug resistance.
While the study explores an interesting concept, several critical issues undermine its scientific rigor and credibility. The primary concerns include inconsistencies in cell line selection, a lack of fundamental understanding regarding Bafilomycin A1’s biological properties, and questionable claims regarding the intracellular trafficking of HER2, a transmembrane protein. While scientific novelty is welcome, the author presents results that are entirely discordant with established literature, significantly compromising the study’s reliability. Given these substantial inconsistencies, I regret to conclude that the manuscript, in its current form, is not suitable for publication.
- Figure 1 One of the most significant methodological concerns arises from the choice of the JIMT1 cell line, which is derived from a HER2-overexpressing breast cancer patient but is well-documented to exhibit high resistance to Trastuzumab. However, the results presented in Figure 1 indicate that Trastuzumab reduces cell viability by 50% within a mere 3-hour incubation period. To my knowledge, no anticancer drug exhibits such a rapid and profound cytotoxic effect within this timeframe. Even if the reported response were valid, the impracticality of such an acute effect raises major concerns regarding potential toxicity and side effects in clinical applications. The experimental design is fundamentally flawed, and the author’s interpretation suggests a lack of foundational knowledge in cell-based assays.
- Figure 1 (continued) The manuscript asserts that natural killer (NK) cells were co-cultured with JIMT1 cells, yet the data fail to provide any evidence supporting their presence or interaction. If the author aims to substantiate this claim, it is imperative to explicitly demonstrate NK cell presence, their viability, and their contribution to the observed effects. Moreover, the rapid onset of the described effects further questions the experimental rigor and reproducibility.
- Figure 2 The manuscript attributes cell death to apoptosis but does not adequately distinguish this process from other forms of programmed cell death, such as autophagy. Notably, autophagy does not involve alterations in intracellular DNA integrity, yet the data do not differentiate these distinct mechanisms. Additionally, the statistical analyses appear questionable, as Figure 2B displays highly significant p-values on both the left and right panels. The author should clarify which statistical software and methodology were employed to derive these values, as the results appear suspect.
- Figure 3 Bafilomycin A1 is a well-characterized inhibitor of vacuolar-type H⁺-ATPase (V-ATPase), known to disrupt intracellular pH homeostasis and inhibit autophagy by preventing lysosomal acidification. Despite the extensive literature on this topic, the manuscript fails to include any analysis of key autophagy-related regulatory proteins, such as ATG family proteins. Instead, the author exclusively presents data on apoptosis-related proteins, which is an incongruous omission given the study’s premise. The lack of appropriate mechanistic validation severely undermines the manuscript’s credibility.
- Figure 5 The rationale for investigating STAT1, CD16, PD-1, and CD95L is poorly articulated. The manuscript provides inadequate background and fails to establish a logical connection between these markers and the study’s central hypothesis. The abrupt introduction of these markers lacks contextual coherence, resulting in an unwarranted leap in logic.
- Figure 6 HER2 is a receptor tyrosine kinase, a transmembrane protein that predominantly resides on the cell membrane. The claim that HER2 enters the cytoplasm upon binding to Bafilomycin A1 is highly speculative and lacks supporting evidence. If the author proposes this mechanism, rigorous experimental validation is required. Specifically, the manuscript should elucidate how HER2 internalization is mediated, detailing potential interactions with membrane-associated proteins, whether HER2 forms dimers or trimers, and the involvement of vesicular transport mechanisms such as clathrin-coated vesicles or endosomes. Without these critical mechanistic insights, the conclusions drawn remain unsubstantiated.
Final Recommendation: While the study explores a potentially interesting hypothesis, the numerous methodological and conceptual flaws render it unsuitable for publication in its current form. The manuscript requires substantial revision, including a reconsideration of cell line selection, robust experimental validation, and a more rigorous mechanistic analysis to support the proposed claims. I encourage the author to address these critical concerns in future submissions to improve the scientific integrity and impact of their work.
Reviewer 2 Report
Comments and Suggestions for Authors
In the manuscript entitled “The autophagy inhibitor bafilomycin inhibits antibody-dependent natural killer cell-mediated killing of breast carcinoma cells,” the authors present several important results. Despite the manuscript being well-prepared, there are some points that must be addressed before recommending its acceptance for publication in the International Journal of Molecular Sciences:
- There are some unclear statements in the introduction. For example, the authors state, “This tumor starts in the breast's milk ducts or lobular cells…” However, malignant lesions are not solely associated with calcifications. Such statements should be revised to avoid providing imprecise information to the reader.
- To emphasize the importance of antibody-dependent cellular cytotoxicity and its role in the immune response to cancer, I suggest including a paragraph discussing recent scientific research in this area, particularly studies using different technologies. A suggested reference is: Guzmán-Mendoza, J.J., et al., Noncytotoxic Carbon Nanotubes Bioconjugated with Fucosyltransferase 4-Derived Peptides Modulate Macrophage Polarization In Vitro. BioNanoSci. 14, 299–317 (2024).
- The scale bars in Figures 1, 2, and 6 are not easily visible. I recommend revising these figures to enhance their readability.
- In Figure 5, the significance of the data related to CD16, PD-1, and CD95L is questionable. The authors should either repeat the experiments or justify the high standard deviations observed in the graphs. The same issue applies to the results presented in Figure 6(c).
- The conclusions of the study should be explicitly stated.
- I suggest including a paragraph before Section 4 highlighting the novelty of this work compared to similar studies already reported in the literature.
- Additionally, I recommend including a paragraph before Section 4 discussing the potential applications derived from this research. This would be of great interest to readers, particularly regarding the future use of the proposed findings.
Round 2
Reviewer 1 Report
Comments and Suggestions for Authors
Dear Editor-in-Chief,
Thank you for considering me as a reviewer for your distinguished journal. Regrettably, the research paper assigned to me appears to diverge significantly from my area of expertise. Consequently, I believe that enlisting a reviewer with a more pertinent background would yield more objective and insightful feedback. My involvement could inadvertently hinder the paper's progression and impact. Therefore, I must respectfully request to be excluded from this review.
However, I am eager to contribute meaningfully in the future should a more suitable opportunity arise. I extend my best wishes for the continued success of both you and your journal.
Sincerely,
Author Response
Response to the reviewer
We appreciate the reviewer's evaluation of our manuscript. The suggested experiments have significantly improved our paper, providing a more comprehensive understanding of our findings and their implications in the field.